# Neural Network-Based Prediction for Secret Key Rate of Underwater Continuous-Variable Quantum Key Distribution through a Seawater Channel

**DOI:** 10.3390/e25060937

**Published:** 2023-06-14

**Authors:** Yun Mao, Yiwu Zhu, Hui Hu, Gaofeng Luo, Jinguang Wang, Yijun Wang, Ying Guo

**Affiliations:** 1School of Information Engineering, Shaoyang University, Shaoyang 422000, China; maoyun3106@csu.edu.cn; 2School of Automation, Central South University, Changsha 410083, China; zhuyw1@jiachengtech.com (Y.Z.); huh@jiachengtech.com (H.H.); wangjinguang@sdwm.edu.cn (J.W.); xxywyj@csu.edu.cn (Y.W.); 3School of Computer Science, Beijing University of Posts and Telecommunications, Beijing 100876, China

**Keywords:** continuous-variable, quantum key distribution, neural network, underwater channel

## Abstract

Continuous-variable quantum key distribution (CVQKD) plays an important role in quantum communications, because of its compatible setup for optical implementation with low cost. For this paper, we considered a neural network approach to predicting the secret key rate of CVQKD with discrete modulation (DM) through an underwater channel. A long-short-term-memory-(LSTM)-based neural network (NN) model was employed, in order to demonstrate performance improvement when taking into account the secret key rate. The numerical simulations showed that the lower bound of the secret key rate could be achieved for a finite-size analysis, where the LSTM-based neural network (NN) was much better than that of the backward-propagation-(BP)-based neural network (NN). This approach helped to realize the fast derivation of the secret key rate of CVQKD through an underwater channel, indicating that it can be used for improving performance in practical quantum communications.

## 1. Introduction

Information security has always been important. Current cryptosystems are dependent on mathematical puzzles without rigorous proofs, and with ever-increasing computing power, such cryptosystems are in danger of being violently broken. In this context, quantum communications have been suggested [1,2], which involve an important technical aspect called a quantum key distribution (QKD) [3,4,5]. Currently, security analysis of QKD protocols involves finding the low bound on the secret key rate: this task, which is usually complex and tedious, may apply to the specific protocols, whereas the achievable bound on the secret key rate is usually not compact enough. Recent results have demonstrated a reliable numerical method, in a finite dimensional environment, for deriving secret key rates [6,7], which depends on solving a convex optimization problem; however, when the Hilbert space in which the bipartite state is located becomes infinite-dimensional, this numerical method cannot be used efficiently.

QKD in discrete variables (DV) [8,9] can be reduced to finite dimensionality by using compressed mappings [10,11,12] or tagged state squarers [13]; however, for QKD in continuous variables (CV) [14,15,16,17], there is no accurate compression model to reduce the dimensionality. Therefore, we have to devote ourselves to finding an effective approach to achieving a compact bound of the secret key rate of the CVQKD system.

CVQKD has unique advantages, such as high-rate modulations with large capacity, which involve Gaussian modulation (GM) and discrete modulation (DM), for processing signals in optical communications. Compared to the GM scheme, the DM scheme has received increasing attention, because of its low cost in experiments. Recent works have focused on asymptotic security proofs of DM-CVQKD with an arbitrary number of the modulated states [14,15,16,17]. The results introduced an assumption of photon number cutoff, in order to reduce the dimension of Hilbert space; therefore, it cannot be said that this approach is a completely strict security proof, and the photon number cutoff assumption cannot be justified. A dimensionality reduction algorithm, without using the photon number cutoff assumption, has been proposed recently [18], which approximates an infinite-dimensional optimization problem, by converting it to a convex optimization problem in a finite-dimensional subspace.

In recent decades, CVQKD has been designed for free-space (FS) communications [19]. Several kinds of CVQKD protocols have been suggested for FS channels, such as satellite-to-satellite links, satellite-to-ground links, air-to-water channels, and so on [20,21]. Unfortunately, the transmission coefficient fluctuates, due to the effects of turbulence in the FS channels, where coherent detection may be distorted, leading to the decreased performance of the quantum communication system. However, there have been a few studies on CVQKD through an underwater channel, where the transmission distance was decreased destructively due to the effect of the noise of the rapidly changing conditions in the seawater. In order to characterize a practical CVQKD through an underwater channel, it is necessary to counteract the effect of excess noise, for data post-processing. Machine learning (ML) has been applied in various fields [22,23,24]. An advantage of the ML-based method is that it consumes less time and resources, yet achieves remarkable results. In this paper, we realized a fast prediction of the secret key rate of the DM-CVQKD, by using an LSTM-based NN model [25], which was based on Bayes optimization for the data post-processing at the receiver. Moreover, prediction of the secret key rate could be achieved for the underwater channels.

The contribution of this work was to predicate the lower bound of the secret key rate, by using an NN model, which involved the LSTM-based NN and the BP-based NN, concerning the dimension-reduced algorithm. We demonstrated that the performance of the LSTM-based NN model was better than that of the BP-based NN model for CVQKD through an underwater channel. These models could speed up the derivation of the secret key rate, compared to the direct numerical method, from which we could obtain a reliable and compact bound of the secret key rate in infinite-dimensional Hilbert spaces.

This paper is organized as follows. In Section 2, we describe our DM-CVQKD through an underwater channel. In addition, we describe how our NN-based scheme for performance prediction of DM-CVQKD was designed. In Section 3, our security analysis with numerical simulation is shown. In Section 4, a summary of the work is provided.

## 2. DM-CVQKD through an Underwater Channel

### 2.1. Description of DM-CVQKD Protocol

The underwater CVQKD system is shown in Figure 1a. Usually, the CVQKD protocol can be described with a prepare-to-measure (PM) scheme, typically known as prepare and measure, where Alice prepares for the DM signals, and sends them to Bob, who performs the measurement operation, and then determines the final secret key by exchanging information over the public channel. In what follows, we detail the steps of CVQKD with the PM scheme:

1. Preparation: For each round, Alice randomly prepares for one of the four quantum states $\left\{|\alpha e^{i(2k+1)\pi /4}\rangle :k\in \left\{1,2,3,4\right\}\right\}$ with equal probability, and sends it to Bob, where α is the amplitude of the quantum state;

2. Measurement: After receiving the signal state, Bob randomly selects a product value from $\left\{\hat{q},\hat{p}\right\}$, to perform homodyne detection, in order to obtain the measurement result, where $\hat{q}$ corresponds to the real part of the quantum state, and $\hat{p}$ corresponds to the imaginary part of the quantum state;

3. Publication and parameter estimation: Alice and Bob exchange information through an authenticated public channel. Bob publishes his chosen summation values for each round through the public channel, and then both parties choose a part of the rounds for parameter estimation, which part of the rounds Alice discloses the quantum states she sends, and Bob discloses the measurements. Based on the public information, both parties can derive the secret key rate under reverse reconciliation (RR). If a secret key rate is not available, both parties terminate the agreement; otherwise, they proceed to the next step;

4. Reverse reconciliation: After the previous steps, the communicating parties use the undisclosed rounds to extract the original key. The specific practice is that both communicating parties follow the same rule for key mapping, and Alice extracts the key according to Bob’s public summation value, which we call reverse reconciliation;

5. Error correction and privacy amplification: in the transmission process of quantum states, the presence of excess noise ξ in the quantum channel makes it inevitable that there are inconsistencies in the interrelated original keys obtained by the two communicating parties. The error correction process is the use of error correction codes by both parties to correct the incomplete agreement bare code, so as to obtain a set of identical binary bits of data. The two communicating parties then have an identical set of binary bits. Unfortunately, Eve, the eavesdropper, may eavesdrop on a set of data sequences that will contain some information about the key; therefore, Alice and Bob choose the appropriate method for private amplification, to generate the final key.

### 2.2. An NN Model for Data Post-Processing

Neural networks are capable of approximating a bounded continuous mapping in a given region [26]. The network model obtained through data training can learn the mapping relationship between input and output, leading to the achievable secret key rate quickly without going through a time-consuming optimization process. The more data that needs to be predicted, the higher the speedup effect is.

Without loss of generality, we suggest an LSTM-based NN model that has an input layer, an output layer, and multiple hidden layers. In addition, this model comes with a Bayes optimization module, as shown in Figure 1b [27,28], which automatically optimizes the hyperparameters based on the training effect, so that we do not need to manually adjust the hyperparameters to show the training effect on the performance of the system.

The LSTM-based NN can solve the gradient explosion or disappearance problem of simple recurrent neural networks [20], and it can update the network model when the received data are added. The main idea of the LSTM is the use of a cell (Figure 1c), which represents the state of a memory unit $\tilde{c}$, as shown in Figure 2.

The data for our training model came from the simulation program of the downscaling algorithm. The distance range [0, 6 m] was varied by a step size of 0.5 m, the depth range [0, 100 m] by a step size of 20 m, the amplitude range [0.6, 0.7] by a step size of 0.02, and the over-noise range [0.001, 0.04] by a step size of 0.001. As the scheme was sensitive to over-noise, the sampling step size for over-noise was small. After sampling by the dimensionality reduction algorithm, we obtained the dataset. Performing the Bayes optimization, the prediction results could be achieved after training the network with the above dataset. When the prediction error was less than zero, i.e., when the predicted value was less than the true value, we considered the prediction result to be secure. The training results for the underwater channel are shown in Figure 3. When the error was less than or equal to 0, the predicted result was secure. The results show that most of the error values were concentrated in the interval [−50%, −10%].

## 3. Security Analysis

### 3.1. Derivation of the Secret Key Rate

The well-known formula for deriving the asymptotic key rate is derived from the difference between the two information-theoretic quantities of private amplification (PA) and error correction (EC) [16,17]. The secret key rate *K* was derived as follows:(1)$$\begin{array}{c} K=\left(\min_{\rho \in S}f\left(\rho \right)\right)-p_{pass}leak_{EC} \end{array},$$
where ρ was the density operator of the quantum states shared by Alice and Bob, *S* was the set of all ρ satisfying the condition known as the feasible set, $p_{pass}$ was the screening probability of each round retained to generate the original key, and $leak_{EC}$ denoted the amount of information leaked in each round of the error correction step.

The first term in the key rate formulation was a convex optimization problem, with ρ as the independent variable. The calculation of the second term could be obtained directly from experimental data [21]. The density operator ρ was an unknown semi-positive definite matrix, but the asymptotic case ρ followed some constraints of the following form:(2)$$\begin{array}{c} Tr\left(\Gamma_{i}\rho \right)=\gamma_{i}, \end{array}$$
where $\Gamma_{i}$ was the ergodic operator, and $\gamma_{i}$ was the expected value of the corresponding ergodic operator. All ρ that satisfied the constraints were expressed as
(3)$$\begin{array}{c} f(\rho )=D(G(\rho )||Z(G(\rho ))) \end{array},$$
where $D(\lambda_{1}\left|\right|\lambda_{2})=Tr\left(\lambda_{1}\log\lambda_{1}\right)-Tr\left(\lambda_{1}\log\lambda_{2}\right)$ was a conditional entropy function, G was a completely positive mapping relation, and Z was a completely positive trace-preserving mapping relation. As both G and Z were linear mappings, and the conditional entropy function was convex, $f\left(\rho \right)$ was a convex function on the feasible set *S*. To extract the secret key rate required finding a ρ that satisfied the constraint, such that $f\left(\rho \right)$ was minimized. The solution to this optimization problem was divided into two steps. The first step found a density matrix $\rho^{\prime }$ that was close or equal to the optimal density matrix $\rho^{*}$, by an iterative algorithm, to obtain $f\left(\rho^{\prime }\right)$ as an upper bound on the key rate. The second step considered the dual problem of the minimization problem, and as the optimal value of the dual problem was less than or equal to the optimal value of the original problem, the optimal value of the dual problem was used as the lower bound of the secret key rate. The closer $\rho^{\prime }$ was to $\rho^{*}$, the closer these two bounds were, and when $\rho^{\prime }$ = $\rho^{*}$, the upper bound coincided with the lower bound.

Each photon received by Bob was different, and could be affected by Eve, so the received photon was in an infinite dimensional Hilbert space, which meant that ρ was infinite-dimensional. Numerical methods can only handle optimization problems where the variables are finite-dimensional, so we had to find a way to make ρ reduce to finite dimensionality. A photon number cutoff assumption was imposed, to achieve the dimensionality reduction of CVQKD, and the basis of Bob’s infinite dimensional Hilbert space was a photon number state $\left\{|n\rangle :n\in N\right\}$, where *N* represented the natural number. This assumption assumed that the number of photons received by Bob was finite, and denoted as $N_{c}$. This assumption truncated the infinite-dimensional Hilbert space, and achieved the dimensionality reduction. The secret key rate obtained was reasonable when a large enough $N_{c}$ was obtained. Improvement of the secret key rate was small when $N_{c}$ was more than 20, as shown in Figure 4.

The imposed photon number cutoff assumption did not constitute a strict security proof; hence, we needed to find an exact security analysis method that eliminated the assumption. In the following, we specify this dimensionality reduction method.

We used $H_{\infty }$ to represent the infinite-dimensional Hilbert space in which ρ resided; $D\left(H_{\infty }\right)$ to represent the normalized density operator on $H_{\infty }$; $\tilde{D}\left(H_{\infty }\right)$ to represent the set of semi-positive definite operators on $D\left(H_{\infty }\right)$; $S_{\infty }$ to represent the feasible set on $\tilde{D}\left(H_{\infty }\right)$; and $\tilde{\rho }$ to represent the density operator on $\tilde{D}\left(H_{\infty }\right)$. Then, the infinite-dimensional optimization problem could be formulated as
(4)$$\begin{array}{c} \min_{\tilde{\rho }\in S_{\infty }}f\left(\tilde{\rho }\right). \end{array}$$
We needed to find a density operator ${\tilde{\rho }}^{\infty }$, to achieve the optimal value, by projecting the infinite-dimensional space onto the finite-dimensional space, to obtain a reduced dimensional representation ${\tilde{\rho }}^{N}$ of ${\tilde{\rho }}^{\infty }$. With $H_{N}$ representing the finite-dimensional Hilbert space on $H_{\infty }$, the semi-positive definite density operator on $H_{N}$ being denoted by $\tilde{D}\left(H_{N}\right)$, and $S_{N}$ denoting the feasible set on $\tilde{D}\left(H_{N}\right)$, the following projection relations were satisfied:(5)$$\begin{array}{c} \Pi \tilde{D}\left(H_{N}\right)\Pi \subseteq \tilde{D}\left(H_{\infty }\right),\;\Pi S_{\infty }\Pi \subseteq S_{N},\;\Pi {\tilde{\rho }}^{\infty }\Pi \subseteq {\tilde{\rho }}^{N}, \end{array}$$
and the finite-dimensional optimization problem was reformulated as
(6)$$\begin{array}{c} \min_{\tilde{\rho }\in S_{N}}f\left(\tilde{\rho }\right), \end{array}$$
where Π was a projection operator. Next, we needed to find ${\tilde{\rho }}^{N}$ that achieved the optimal value of $f(\tilde{\rho })$. As shown by [18], the infinite-dimensional optimization problem was related to the finite-dimensional optimization problem, in that
(7)$$\begin{array}{c} f\left({\tilde{\rho }}^{N}\right)-\Delta \left(W\right)\leq f\left({\tilde{\rho }}^{\infty }\right), \end{array}$$
where Δ(W) was a non-negative correction term that was used to compensate for errors arising from the photon number cutoff assumption, and *W* represented the weight of the key rate bound outside the finite-dimensional subspace. The conditions also required that the projection of $f\left(\tilde{\rho }\right)$ on $S_{\infty }$ was nearly uniformly decreasing [18], satisfying $Tr\left({\tilde{\rho }}^{\infty }\right)\leq W$. Therefore, we needed to determine four components in Equation (Equation 7): finite-dimensional subspace $H_{N}$; finite-dimensional feasible set $S_{N}$; weights W outside the subspace W; and correction term Δ.

By this stage, we had obtained all the components of the infinite-dimensional optimization problem; hence, the secret key rate under the infinite-dimensional space could be derived as
(8)$$\begin{array}{c} K_{\infty }=\left(\min_{\rho \in S_{N}}f\left(\tilde{\rho }\right)\right)-leak_{EC}-\Delta \left(W\right). \end{array}$$

### 3.2. Effects of Excess Noise

In order to show the performance of the DM-CVQKD, we describe the characteristics of the underwater channel. Then, we show the effects of excess noise on the secret key rate.

In what follows, we demonstrate the transmission rate in the underwater channel. The calculation of the transmission rate of the underwater channel was complicated, involving water type and chlorophyll content, as shown in Appendix A. The attenuation rate was high in the underwater channel. We considered the Monte-Carlo model [19], where the communication light wavelength is 520 nm, and the water type is pure seawater. The transmission rate *T* of the underwater channel was related to the absorption coefficient *a* and the scattering coefficient *b*, depending on transmission distance and depth. Then, we had
(9)$$\begin{array}{c} T=e^{-cL}, \end{array}$$
where *c* was a constant that involved the sum of *a* and *b* related to the depth.

Next, we considered the excess noise in the underwater channel. In Figure 5 and Figure 6, we show the effects of excess noise ξ on the secret key rate. We took a step size of 0.005, and simulated in the interval [0.005, 0.04] for excess noise. The result shows that the secret key rate decreased as the excess noise decreased gradually.

Moreover, the numerical simulations showed that the underwater DM-CVQKD system was sensitive to excess noise, and that the transmission distance decreased as the excess noise increased. When the excess noise reached 0.04, the maximum transmission distance in the underwater channel was less than 0.5 m.

### 3.3. Post-Selection

Alice and Bob were able to use post-selection for data reconciliation, filtering out unqualified data, so as to improve reconciliation efficiency and tolerance to excess noise, resulting in an increased secret key rate. When enabling the post-selection or not, we set the given post-selection parameter Δ to zero or greater than zero. As shown in Figure 6, we considered numerical simulations for types of excess noises.

We found that post-selection in CVQKD reduced the error rate, by discarding the results near zero: this was why, as the excess noise increased, erroneous results were more likely to occur around the zero point [21]. The numerical simulations showed that the post-selection operation improved the protocol key rate and the tolerance to noise. For the small excess noises, such as ξ=0.01 and ξ=0.02, the post-selection-involved improvement seemed small; when the excess noise reached ξ=0.03, the post-selection was an obvious improvement on the secret key rate; therefore, the post-selection was necessary when the excess noise underwater became high.

### 3.4. Simulation Results

In Figure 7, we show the prediction of the secret key rate of the CVQKD system in an underwater channel. The parameter settings in numerical simulations are shown in Table 1.

In order to demonstrate the advantage of the LSTM-based NN on performance improvement, we compared the prediction results of the LSTM-based NN and the BP-based NN to the traditional CVQKD, without involving the NN model. In the numerical simulations, we set excess noise ξ=0.01, post-selection Δ=0, reconciliation efficiency β=0.95, and modulation amplitude α=0.66, respectively. At the transmission distance of 0.5 m and 7 m, the prediction results of the BP-based NN and the LSTM-based NN improved by about 1.5% and 5.5%, respectively. According to the simulation results, both the BP-based NN and the LSTM-based NN showed performance improvement of the secret key rate, whereas the LSTM-based NN resulted in a higher secret key rate, compared to the BP-based NN.

## 4. Conclusions

We propose an NN approach to predicting the achievable secret key rate of the DM-CVQKD system through an underwater channel. The secret key rate of the CVQKD system can be improved when NN-based data post-processing is used for the receiver. In addition, the prediction performance of the LSTM-based NN model performs better than that of the BP-based NN model for the CVQKD. The numerical simulations show that the LSTM-based NN model can improve prediction accuracy compared to the BP-based NN model. Our approach paves the way for predicting the performance of the CVQKD system.

## Figures and Tables

**Figure 1 entropy-25-00937-f001:**
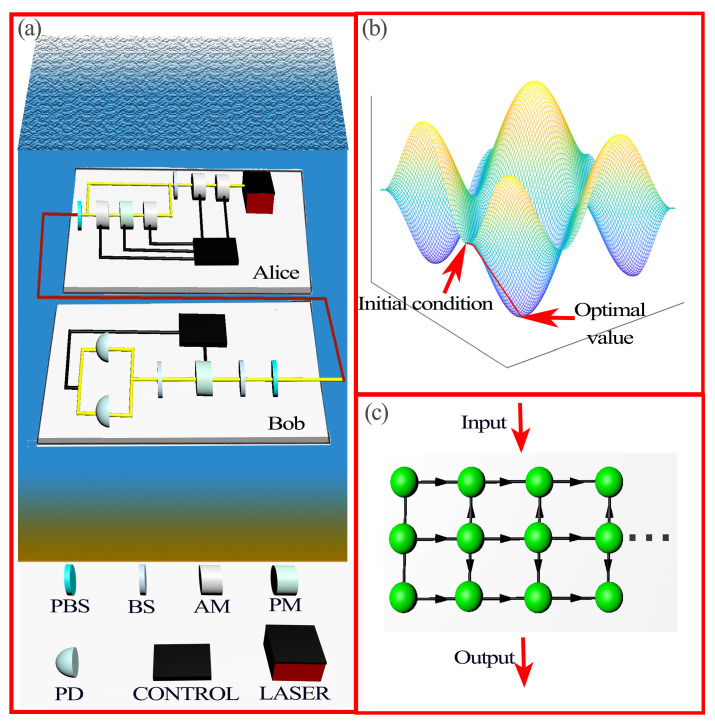
Scenario diagram of DM-CVQKD through an underwater channel: (**a**) Underwater environment. PBS: Polarization Beam Splitter, BS: Beam Splitter, AM: Amplitude Modulator, PM: Phase Modulator, PD: Photo Diode; (**b**) Bayesian optimization; (**c**) The trained neural network.

**Figure 2 entropy-25-00937-f002:**
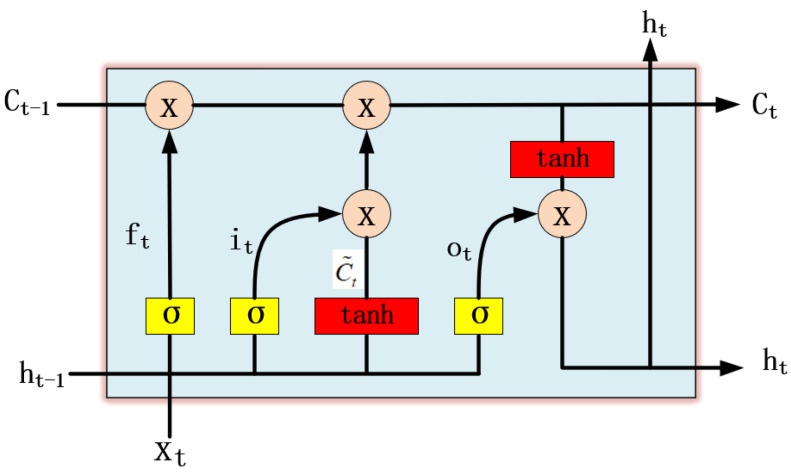
The internal structure of a cell of the LSTM-based NN. The internal state $C_{t-1}$ of the previous moment, the external state $h_{t-1}$, and the network input $x_{t}$ of the current moment are used as the input of the cell, and the current internal state $C_{t}$ and external state $h_{t}$ are obtained as the output of the cell by gate operations, i.e., forget gate, input gate, and output gate, respectively.

**Figure 3 entropy-25-00937-f003:**
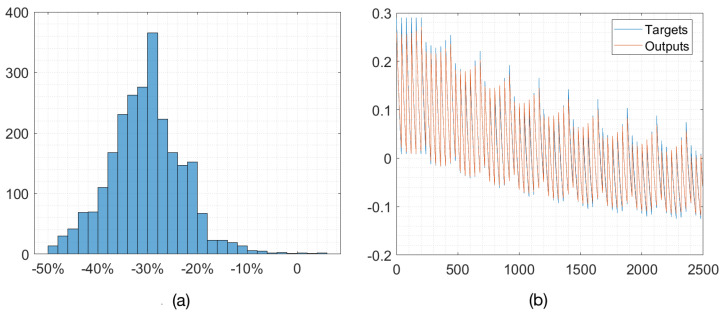
Training results of the LSTM-based NN: (**a**) Error histogram of prediction. The number of samples vs relative error of the train dataset; (**b**) The training set predicted and expected values. The red line is the predicted value, and the blue line is the expected value.

**Figure 4 entropy-25-00937-f004:**
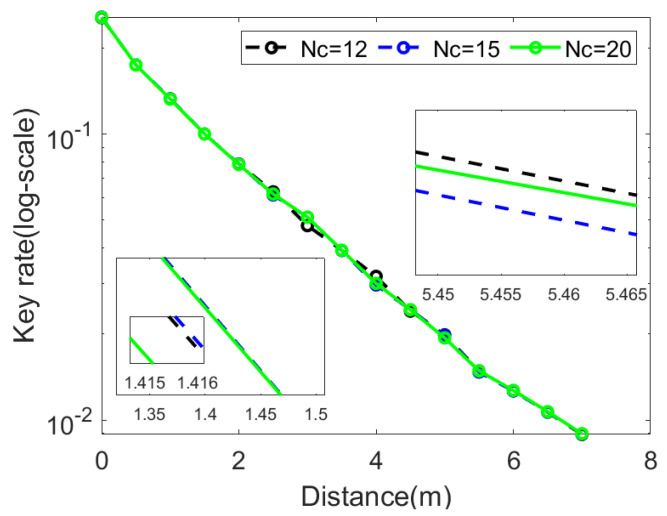
The secret key rate as a function of the transmission distance for the given photon cutoff numbers. The black dashed line is $N_{c}=12$, the blue dashed line is $N_{c}=15$, and the green line is $N_{c}=20$. The key rate float is about 0.55% for $N_{c}$ from 12 to 15, and 0.2% for $N_{c}$ from 15 to 20. For the increased $N_{c}$, the secret key rate is not obviously improved, but the computation time increases significantly.

**Figure 5 entropy-25-00937-f005:**
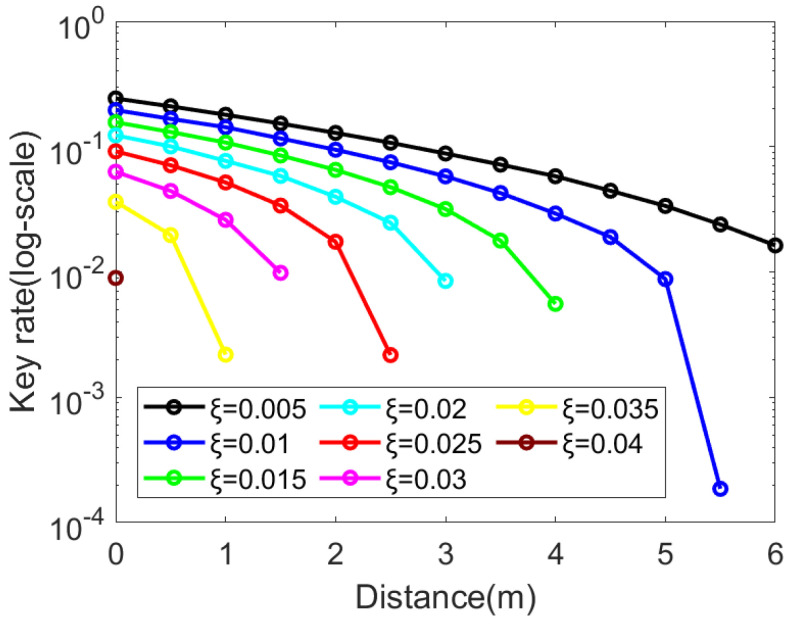
Effects of excess noise ξ on the secret key rate. The lines from top to bottom indicate the excess noise ξ∈{0.005,0.01,0.015,0.02,0.025,0.03,0.035,0.04}. We set the amplitude α=0.66, post-selection parameter Δ=0, depth = 100 m, and reconciliation efficiency β=0.95.

**Figure 6 entropy-25-00937-f006:**
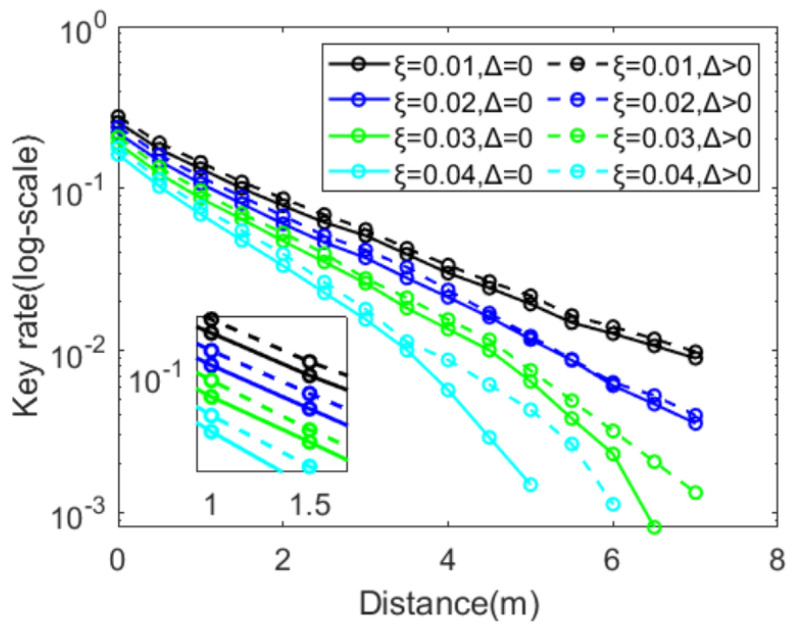
Effects of excess noise ξ on the secret key rate with the given post-selection. The solid line indicates Δ=0, and the dashed line indicates Δ>0.

**Figure 7 entropy-25-00937-f007:**
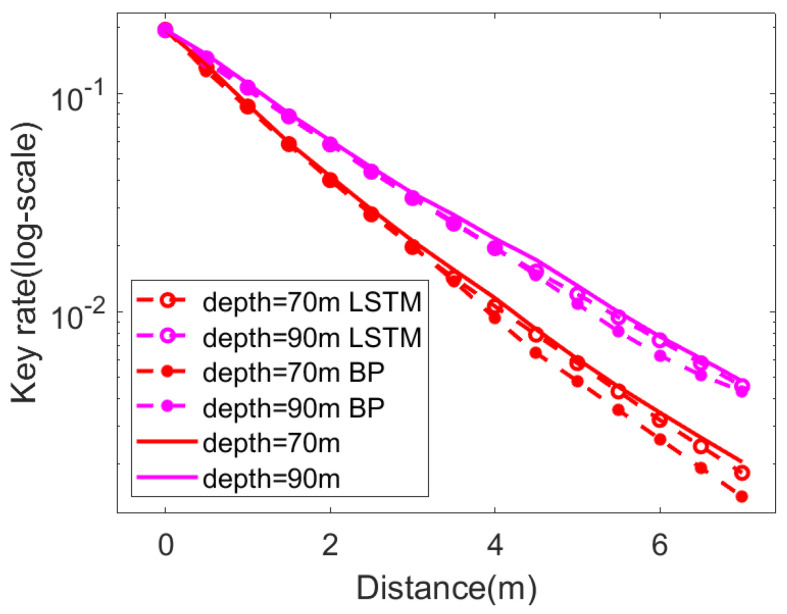
Prediction results of the NN-based CVQKD. Solid lines represent the initial value with photon cutoff method, the hollow dotted line represents LSTM-based NN, and the solid dotted line represents BP-based NN. The pink line represents a depth of 70 m, and the red line represents a depth of 90 m.

**Table 1 entropy-25-00937-t001:** Parameter Setting.

Symbols	Value	Description
α	0.6–0.7	Amplitude
$\hat{q}$	–	Orthogonal amplitude components
$\hat{p}$	–	Orthogonal phase component
$N_{c}$	20	Photon cutoff number
ξ	0–0.04	Excess noise
Δ	0.01	Post-selection parameter
β	0.95	Reconciliation efficiency

## Data Availability

Data available on request from the authors.

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
