# Peer review of "Neural Network-Based Prediction for Secret Key Rate of Underwater Continuous-Variable Quantum Key Distribution through a Seawater Channel"

_entropy, 2023, doi:10.3390/e25060937_

Round 1
Reviewer 1 Report
The topic of the paper is interesting, and also the idea of using AI to predict the key-rate performance could be promising.
However, the overall quality of the paper is low, and the amount of information provided to the readers is low. The writing style is poor, and in many parts (starting from the abstract), the meaning of some sentences is hard to understand.
The application of the adopted Neural Network (NN) to the considered problem is not clear. The authors report a very high-level description of the used algorithms (LSTM,BP), but do not provide any detail on many important aspects, such as NN training, testing etc. Also, just one simulation (summarized by Fig.7) shows the use of the NN to predict the key rate.
Numerical results are about the effect of noise on the key-rate, but the analysis does not provide any meaningful insight into the QKD exchange mechanism, besides the quite expected result that noise can affect the performance.
In the simulations, what is the difference between "distance" and "depth"? Do the author mean that both TX and RX are at a depth of, e.g., 80 meters, at a horizontal distance of 6 m? If such is the case, how does depth affect the key-transmission?
When dealing with underwater QKD, the optical properties of water play a fundamental role in the performance. In this paper, only the effect of "excess noise" has been considered. This approach is not suitable for a publication on a journal which aims at a audience of experts on Underwater QKD.
Some concepts are introduced without any explanation or proper referencing. For instance, the paper starts discussing about the post-selection parameter in the caption of Fig.6, without any introduction.
Overall, the paper is not suitable for publication on a journal, without significant and deep changes.
The writing style is not adequate for a journal paper. Starting from the abstract, many sentences are hard to understand for the reader.
For instance, from the abstract:
"The long short-trem memory (LSTM)-based neural network (NN) model is elegantly employed in order to speed up the derivation of the secret key rate, resulting in the compact bound. The numerical simulations show that the bound of the secret key rate can be achieved, which performs better than that of the backward propagation (BP)-based neural network."
If the bound is not obtained in closed form, how can it be compact? In my understanding, it is provided as an output by the NN. Also, if the NN is trained with the results of simulations, there is no surprise in the fact that the bound can be achieved.
In the paper, there are also many typos or trivial grammar errors which render the paper very confused. Just as another example, in the conclusions:
"The performance of the NN-based CVQKD system can be improved in terms of the secret key rate."
This sentence, as many others throughout the entire manuscript, render the paper really hard to understand to readers.
Reviewer 2 Report
1. The introduction requires rewriting as it doesn't do its job of motivating this work. In particular I believe due to the mention of "lower bound" in line 21, and the overall context of that paragraph and subsequent one that this refers to the finite size effect of CVQKD protocols but this is not elaborated on. Please motivate why an underwater channel.
2.Please elaborate on "simple and stable modulations"
3.Line 32, a GM CVQKD setup should have exactly the same electroptic complexity as a DM CVQKD setup, the difference being the protocols involved, and limitations then of limited DAC/ADC resolution and the security epsilons involved.
4. Line 44, is the guarantee of security from the protocol involved or the ML method?
5. Line 65, typically called prepare and measure.
6. Fig 1, what is the role of the PBS? Given that this work uses essentially QPSK, why is there a need for such a complicated transmitter structure? The authors introduced this CVQKD system as being a simple experiment yet there are 5! modulators just at the Tx. (c) shows nothing about the neural network other than that there's 3 layers... and no other details, please revise. With the PBS at the receiver, is there an expected effect of polarization rotation in the simulations, should this be a polarization controller? is this just being used as a polarizer? because there's only one input and one output. I assume the PBS at the Tx is being used as a polarization beam combiner.What is the role of beamsplitters (BS?) in the Rx. Are they VOAs? and beamsplitters being used as representations for the theoretical model?
7. Line 95, Eve can eavesdrop at any point in the quantum channel given that its untrusted and the communication over the classical channel is completely available to her... this is misconstrued in this section.
8. Please provide more detail on the simulations used e.g. symbols used, modulation variance, channel loss.
9. Please explain why your method achieves an average prediction inaccuracy of a 30% reduction in key rate, was this an intended metric to attempt to force always a prediction less than the convex optimization method?
10. Figure 5. Is tehre any other impairments in this simulation such as laser phase noise? How is the excess noise simulated? via additive noise at the receiver?
11. Consider using different symbols per set of results or a color-blind friendly colormap.
12. Results figures would be much more readable with grid lines.
13. Section 2 should have the details on the underwater channel transmittivity. This section is currently a general explanation of any CVQKD system modulating QPSK employing reverse reconciliation. a and b parameters should be clearly stated in simulations.
14. Would the author's NN method perform equally well on the photon cutoff procedure vs their proposed one? How does the author's problem formation compare to photon cutoff method? Has it been presented elsewhere and simply needs citation?
15. Could the authors comment on the comparative processing time between the three methods discussed in this work.
Overall there is a considerable lack of clarity in this manuscript, it requires significant rewriting. E.g. "simple experiment" in the abstract. There are CV-QKD systems that have a complex implementation, particularly with squeezed state for example, or time multiplexed reference pulses. The authors should carefully review the language used.
The title is in particular quite confusing since this work uses a NN to estimate the secret key rate but the CVQKD protocol is not based on it. It is a discrete modulation protocol.
This paper would probably be best split up into its theoretical and machine learning parts to better focus on them.
Reviewer 3 Report
The authors proposed the implementation of discrete modulation continuous-variable quantum key distribution (DM-CVQKD) and employed a machine learning-based neural network (NN) technique to enhance its performance to predict the secret key rate and maximal transmission distance. Specifically, they used an effective optimization approach called the LSTM-based NN model to derive a compact bound of the secret key rate of the CVQKD system. In addition, they also compared two approaches LSTM-embedded NN and the BP-embedded NN for DM-CVQKD through the underwater channels and found the performance of LSTM works better and faster. Their DM-CVQKD protocol is written clearly, along with the security analysis including the dimensionality reduction method. They have also discussed the effects of excess noise on the secret key rate in underwater channel. The study seems complete and impressive. Therefore, I accept the manuscript after a minor revision as follows.
1. Can the authors cite the exact reference they talk about in line 34—“Recent works have focused on asymptotic security proofs of …”.
2. Can the authors elaborate on more on line 49-- to make it more clear on how “…..can be automatically constructed in underwater channel”.
Minor editing of the English language required
Round 2
Reviewer 1 Report
The writing style should be further improved.
The writing style should be further improved. Some parts of the paper still need improvement.
Author Response
We deeply appreciate the review of our manuscript, and we have revised the the writing style of the previous version of the manuscript. Thanks very much for your kindly suggestions for the improvement of our manuscript.Reviewer 2 Report
I thank the authors for their revisions in the manuscript and for their interesting work. I would suggest maybe a citation or two if appropriate for Appendix A as well as grid lines in figures 4-7 as well as an order of magnitude estimate on the computation time decrease instead of merely "significant" for precision.
Line 35 is incorrect, CV QKD can encode information onto any properties of light that are continuous... we use amplitude and phase because it is convenient but we could encode onto continuous polarization states if we wanted to.
37. It is enough to state that a GM scheme encodes data pulled from a Gaussian distribution... amplitude has nothing to do with it, you could encode a Gaussian distribution onto phase for example.
Thank you for making the manuscript much more readible.
Minor points:
Abstract is still a bit confusing in line 7-8 with respect to "made much compact for..."
Check if there are no more typos such as on line 57, access -> excess
Line 66 - lower
